# Meta-analysis of the effectiveness and safety of Shenyankangfu tablets combined with losartan potassium in the treatment of chronic glomerulonephritis

**Pan-Feng Feng[1], Xiang-Fan Chen[1], Nan Sheng[2], Long-Xun Zhu** [ID][1]*

**1** Department of Pharmacy, Affiliated Hospital 2 of Nantong University, and First People's Hospital of Nantong City, Nantong, Jiangsu Province, China, **2** Clinical Medical Research Center, Affiliated Hospital 2 of Nantong University, and First People's Hospital of Nantong City, Nantong, Jiangsu Province, China

* 948017729@qq.com

## Abstract

### Objective

To conduct a systematic review of the efficacy and safety of Shenyankangfu tablets in combination with losartan potassium in the treatment of chronic glomerulonephritis.

### Method

We searched PubMed, Embase, Cochrane Library, CNKI, WanFang Data, and WeiPu for comparative studies on Shenyankangfu tablets in combination with losartan potassium in the treatment of chronic glomerulonephritis. The search period runs from the establishment of the database until September 2021. Data extraction and quality evaluation were carried out on the documents that met the inclusion criteria, and a meta-analysis of the included literature was conducted using the RevMan5.3 software.

### Results

A total of 17 randomized controlled trials that met the inclusion criteria were included, with a total sample size of 1680 patients (841 patients in the study group and 839 in the control group). The effective rate was significantly higher in the study group than in the control group [RR = 1.22, 95% CI (1.16, 1.27), P < 0.00001]. In addition, 24-hour urine protein levels [SMD = -1.11, 95% CI (-1.40, -0.83), P < 0.00001], urine NAG enzyme [SMD = -0.99, 95% CI (-1.27, -0.72), P < 0.00001], leukotactin-1 [SMD = -2.43, 95% CI (-3.50, -1.35), P < 0.00001], and the incidence of adverse reactions [RR = 0.43, 95% CI (0.28, 0.66), P < 0.00001] were lower in the study group when compared to the control group.

### Conclusion

It is safer to treat chronic glomerulonephritis with Shyenyankangfu tablets in combination with losartan potassium. At the same time, it alleviates disease-related symptoms, reduces the influence of cytokine levels, and has fewer adverse reactions, making it more conducive

**Data Availability Statement:** All relevant data are within the paper and its Supporting Information files.

**Funding:** This work was supported by Jiangsu Pharmaceutical Association-HengRui Hospital Pharmacy Fund (No. H202047), Nantong Health Commission Fund (No. QA2021007, QA2021006, QA2021014) and Development Fund of KangDa college of Nanjing medical university (No. KD2021KYJJZD127). The funders had no role in study design, data collection and analysis, decision to publish, or preparation of the manuscript.

**Competing interests:** The authors have declared that no competing interests exist.

to disease recovery. However, additional multi-center, randomized, control trials with large sample sizes must be conducted to confirm the findings.

## 1. Introduction

Chronic glomerulonephritis (CGN) is a relatively common primary glomerular disease with a high prevalence [1, 2]. CGN is a primary glomerular disease, mainly due to the influence of various causes with the main clinical manifestations of proteinuria, hematuresis, edema, and hypertension [3, 4]. It has an insidious onset and a delayed course, and the majority of patients progress to chronic renal failure. CGN is characterized by interstitial fibrosis, glomerulosclerosis, and tubular atrophy and is the final common pathway of most glomerular disorders (primary or secondary) when poorly managed or unresponsive to therapies. Since CGN is prone to recurrent episodes, if the disease progresses and is not effectively treated, it can result in life-threatening complications such as renal failure.

Because proteinuria and hypertension are independent risk factors influencing the prognosis of chronic nephritis, controlling them is an important measure to delay renal failure [5]. Currently, clinical treatment methods primarily involve the use of angiotensin II receptor blockers (ARBs) to reduce glomerular intravascular pressure and proteinuria, thereby delaying chronic renal function changes. However, when used alone, ARBs have poor therapeutic effects.

Although studies conducted in China have shown that the combination of traditional Chinese medicine (Shenyankangfu tablets, SYKF tablets) and losartan potassium tablets has a greater clinical benefit than losartan potassium tablets alone in the treatment of CGN, no systematic analysis has yet been conducted in this area. Therefore, we intend to conduct a meta-analysis to investigate the efficacy and safety of the combination of SYKF tablets and losartan potassium tablets in the treatment of CGN.

## 2. Materials and methods

### 2.1 Search strategy

Registration: Based on the Cochrane Handbook criteria and Preferred Reporting Items for System Reviews and Meta-Analyses (PRISMA statement), our meta-analysis was registered with PROSPERO (CRD42022345149). Searcher: The literature search and information extraction were carried out independently by two searchers. Database: PubMed, Embase, Web of Science, CNKI, WanFang database (WANFANG), and VIP database (VIP). Search terms: "Shenyankangfu tablets," "losartan potassium tablets," and "chronic glomerulonephritis." Search time range: From the date of establishment of each database until September 2021.

### 2.2 Inclusion and exclusion criteria

**2.2.1 Inclusion criteria.**   Study type: All prospective randomized controlled trials, whether blinded or not, were included. Study patients: All patients meet the criteria for the diagnosis of chronic nephritis set by the World Health Organization (WHO). Intervention measurements: The control group was treated with SYKF or losartan potassium alone, while the experimental group was treated with the combination of SYKF and losartan potassium (50 mg, Po, qd). Outcome indicators: The primary indicators include the effective rate. The efficacy criteria are as follows: Significantly effective, clinical signs and symptoms are significantly alleviated, 24 h

urine protein quantitative decrease > 40% or urine protein decrease 2 "+", urine sediment count red blood cell decrease > 40% or urine red blood cell decrease 2 "+"; Effective, clinical signs and symptoms are relieved, 24 h urine protein quantitative decrease ≤ 40% or urine protein decrease 1 "+"; Invalid, clinical signs, symptoms, and related indicators aggravated or not alleviated. Total effective rate = Significant effective rate + effective rate. The secondary indicators include serum creatinine (Scr), Blood urea nitrogen (BUN), 24-hour urinary protein quantity, urine NAG enzyme, and LKN-1.

**2.2.2 Exclusion criteria.**   1: Reviews, animal experiments, case reports, and expert experience reports; 2: Repeated publications; 3: Studies where full-text data cannot be extracted for statistical analysis.

## 2.3 Data extraction and bias risk assessment

The literature screening and data extraction were carried out independently by two researchers. A unified extraction table was used for data extraction. Where there was a disagreement, it was resolved through discussion or with the assistance of a third researcher. The bias risk of included studies was assessed according to the Cochran Handbook. The evaluation content includes random sequence generation, allocation concealment, blinding of participants and personnel, blinding of outcome assessment, incomplete outcome data, selective reporting, and other bias. Each item was classified as low, high, or unclear risk.

## 2.4 Statistical analysis

Statistical analysis was performed using RevMan5.3 software. The evaluation indicators of the study are binary variables, such as effective rate and incidence of adverse reactions. Relative risk (Risk ratio, RR) was used as the analysis statistic, and its 95% confidence interval (CI) was calculated. Continuous variables include 24-hour urine protein, urine N-Acetyl-D-glucosamine (NAG) enzyme, blood urea nitrogen, blood creatinine, and leukotactin-1 (LKN-1). The standardized mean difference (SMD) was used as the analysis statistic, and its 95% CI was calculated. A Z-test was used to analyze RR; $P < 0.05$ indicates that the two evaluation indices are statistically different; otherwise, there is no statistical difference. The $\chi^2$ test was used to analyze heterogeneity. If statistical heterogeneity existed between studies ($P < 0.1$, $I^2 > 50\%$), the random effects model was used for analysis; otherwise, the fixed effects model was employed.

# 3. Results

## 3.1 Basic characteristics of included studies

The literature search and screening process is shown in Fig 1. Initially, 63 kinds of related literature were retrieved. After reading the title and abstract and excluding irrelevant or repetitive articles, read the full text and standard evaluation of the remaining 36 kinds of literature. 23 non-core literature in China, drug inconsistencies, data loss, and non-detailed literature were deleted. Finally, 17 kinds of literature (abstract could be obtained in Supplementary material abstract) that met the criteria were included in the meta-analysis [6–22]. A total of 841 patients in the study group and 839 patients in the control group were included. The basic characteristics of each study are summarized in Table 1.

## 3.2 Quality evaluation results of included studies

The articles included are all Chinese literature. There are 12 kinds of literature mentioning specific random methods [6, 9, 11, 13–18, 20–22], and the remaining 5 kinds of literature only mention the use of random grouping [7, 8, 10, 12, 19], with no specific random methods

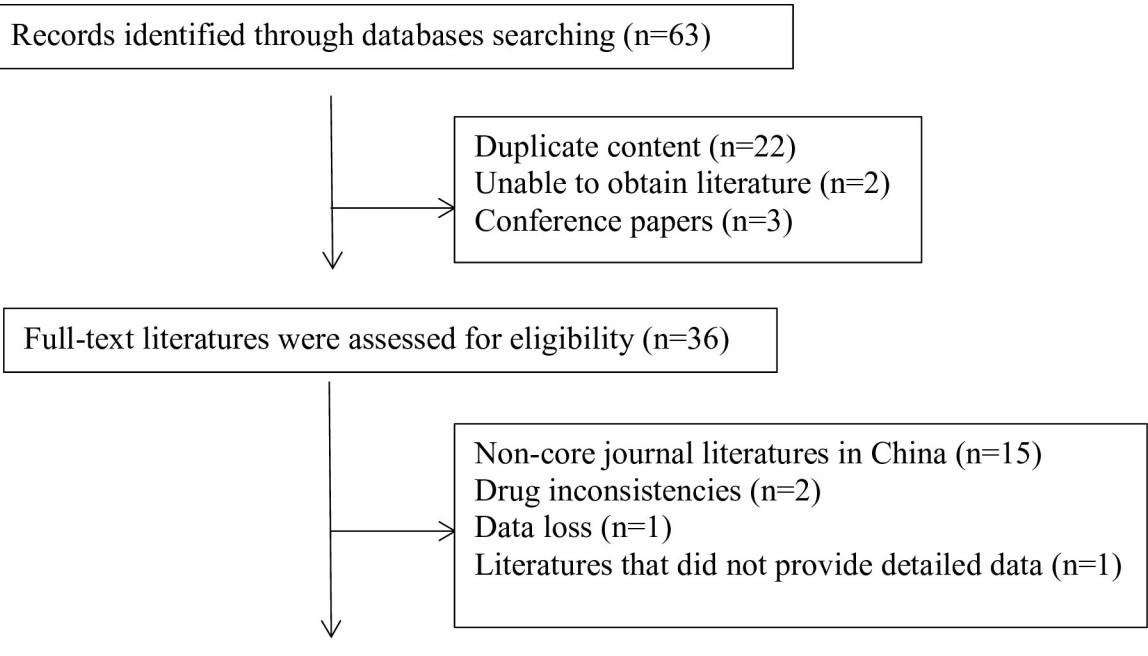

**Fig 1. Summary of the process for identifying candidate studies.**

described. All the reported research data are complete. None of the studies provided information on blinding, sample size estimation, or random allocation concealment. The risk of bias assessment is shown in Fig 2.

### 3.3 Results of Meta-analysis

**3.3.1 Effective rate.** The total effective rate was reported in 15 studies [6, 8–19, 21, 22]. There was no statistical heterogeneity among the studies (P = 0.42, $I^2$ = 3%), and the fixed effect model was used for the meta-analysis. The meta-analysis results revealed that the total effective rate of patients in the study group was significantly higher than that of the control group [RR = 1.22, 95% CI (1.16, 1.27), P < 0.00001]. The results are shown in Fig 3.

**3.3.2 24-h urine protein levels.** 14 studies [6–8, 10–16, 19–22] reported changes in 24-h urine protein levels, and there was statistical heterogeneity between the studies (P < 0.00001, $I^2$ = 84%). The random effects model was used for the meta-analysis. The results are shown in Fig 4 and S1 Table in S1 File. The study group had significantly lower 24-h urine protein levels than the control group [SMD = -1.11, 95% CI (-1.40, -0.83), P < 0.00001].

**3.3.3 Serum creatinine (Scr).** 12 studies [6, 8, 10, 12–17, 19, 21, 22] reported changes in Scr, and statistical heterogeneity (P < 0.00001, $I^2$ = 91%) existed between the studies. The random effects model was used for the meta-analysis. The results are shown in Fig 5 and S2 Table in S1 File. The value of Scr did not change significantly in the study group compared to the control group.

**3.3.4 Blood urea nitrogen (BUN).** 10 studies [6, 8, 10, 13–15, 17, 19, 21, 22] reported changes in blood urea nitrogen, and there was statistical heterogeneity between the studies (P < 0.00001, $I^2$ = 91%). The random effects model was used for meta-analysis. The results are

**Table 1. The characteristics of clinical studies.**

| Study (years) | Number(n = male/female) Age (years) | | Treatments | | Outcome index | Intergroup difference |
|---|---|---|---|---|---|---|
| | **Study group** | **Control group** | **Study group** | **Control group** | | |
| Chen SY 2021 | 40(23/17) 41.4±3.7 | 40(21/19) 41.5 ±3.4 | SYKF tablet (5 tablets, tid) + Losartan Potassium tablet (1 tablets, qd)) | Losartan Potassium tablet (1 tablets, qd) | 1. Effective rate 2. Urine NAG enzyme 3. 24-hour urinary protein quantity 4. Cytokines 5. Adverse Reaction 6. Scr 7. BUN | 1. P<0.05 2. P<0.05 3. P<0.05 4. P<0.05 5. P<0.05 6. P<0.05 7. P<0.05 |
| Guo YM 2018 | 63(35/28) 45.3±8.2 | 63(36/27) 46.5 ±9.5 | SYKF tablet (5 tablets, tid) + Losartan Potassium tablet (1 tablets, qd)) | Losartan Potassium tablet (1 tablets, qd) | 1. 24-hour urinary protein quantity 2. Cytokines 3. T cell subsets | 1. P<0.05 2. P<0.05 3. P<0.05 |
| Li HX 2017 | 120(58/62) 43.9±7.7 | 120(56/64) 43.6±7.5 | SYKF tablet (5 tablets, tid) + Losartan Potassium tablet (1 tablets, qd)) | Losartan Potassium tablet (1 tablets, qd) | 1. Effective rate 2. 24-hour urinary protein quantity 3. Urine NAG enzyme 4. BUN 5. Scr 6. Adverse Reaction | 1. P<0.05 2. P<0.05 3. P<0.05 4. P>0.05 5. P>0.05 6. P<0.05 |
| Li MB 2017 | 34(20/14) 42.6±14.9 | 34(21/13) 42.1 ±15.2 | SYKF tablet (5 tablets, tid) + Losartan Potassium tablet (1 tablets, qd)) | SYKF tablet (5 tablets, tid) | 1. Effective rate 2. Adverse Reaction | 1. P<0.05 2. P<0.05 |
| Li XD 2017 | 34(19/15) 35.0±4.6 | 34(18/16) 34.8 ±5.3 | SYKF tablet (5 tablets, tid) + Losartan Potassium tablet (1 tablets, qd)) | Losartan Potassium tablet (1 tablets, qd) | 1. Effective rate 2. Symptom score 3. 24-hour urinary protein quantity 4. IL-6,8 5. BUN 6. Scr | 1. P<0.05 2. P<0.05 3. P<0.05 4. P<0.05 5. P>0.05 6. P>0.05 |
| Lv TH 2020 | 44(24/20) 50.5±5.3 | 44(23/21) 51.0 ±5.5 | SYKF tablet (5 tablets, tid) + Losartan Potassium tablet (1 tablets, qd)) | Losartan Potassium tablet (1 tablets, qd) | 1. Effective rate 2. 24-hour urinary protein quantity 3. IL-6,8 4. Adverse Reaction | 1. P<0.05 2. P<0.05 3. P<0.05 4. P<0.05 |
| Pan HX 2016 | 50(26/24) 48.5±3.2 | 50(27/23) 47.3 ±3.5 | SYKF tablet (5 tablets, tid) + Losartan Potassium tablet (1 tablets, qd)) | Losartan Potassium tablet (1 tablets, qd) | 1. Effective rate 2. 24-hour urinary protein quantity 3. GFR 4. Adverse Reaction 5. Scr | 1. P<0.05 2. P<0.05 3. P<0.05 4. P<0.05 5. P<0.05 |
| Qiu H 2017 | 60(31/29) 42.9±2.7 | 60(32/28) 43.4 ±3.4 | SYKF tablet (5 tablets, tid) + Losartan Potassium tablet (1 tablets, qd)) | Losartan Potassium tablet (1 tablets, qd) | 1. Effective rate 2. 24-hour urinary protein quantity 3. Urine NAG enzyme 4. Adverse Reaction 5. BUN 6. Scr | 1. P<0.05 2. P<0.05 3. P<0.05 4. P<0.05 5. P>0.05 6. P>0.05 |
| Qiu J 2018 | 30(18/12) 52 ±3 | 30(13/17) 52 ±3 | SYKF tablet (5 tablets, tid) + Losartan Potassium tablet (1 tablets, qd)) | Losartan Potassium tablet (1 tablets, qd) | 1. Effective rate 2. 24-hour urinary protein quantity 3. BUN 4. Scr | 1. P<0.05 2. P<0.05 3. P<0.05 4. P<0.05 |

(*Continued*)

**Table 1.** (Continued)

| Study (years) | Number(n = male/female) Age (years) | | Treatments | | Outcome index | Intergroup difference |
|---|---|---|---|---|---|---|
| | Study group | Control group | Study group | Control group | | |
| Su Y 2019 | 60(45/15) 34.3±3.2 | 60(42/18) 35.6 ±4.1 | SYKF tablet (5 tablets, tid) + Losartan Potassium tablet (1 tablets, qd)) | Losartan Potassium tablet (1 tablets, qd) | 1. Effective rate 2. 24-hour urinary protein quantity 3. BUN 4. Scr 5. GFR | 1. P<0.05 2. P<0.05 3. P<0.05 4. P<0.05 5. P>0.05 |
| Wang XL 2018 | 40(21/19) 49.5±9.4 35.6±9.4 | 40(22/18) 49.7 ±9.8 | SYKF tablet (8 tablets, tid) + Losartan Potassium tablet (1 tablets, qd)) | Losartan Potassium tablet (1 tablets, qd) | 1. Effective rate 2. 24-hour urinary protein quantity 3. Scr 4. LKN-1 5. IL-8 6. TNFα | 1. P<0.05 2. P<0.05 3. P<0.05 4. P<0.05 5. P<0.05 6. P<0.05 |
| Wu Y 2016 | 50(31/19) 54.4±3.2 | 50(33/17) 55.4 ±3.3 | SYKF tablet (5 tablets, tid) + Losartan Potassium tablet (1 tablets, qd)) | Losartan Potassium tablet (1 tablets, qd) | 1. Effective Rate 2. LKN-1 3. IL-33 4. TNFα 5. BUN 6. Scr 7. GFR | 1. P<0.05 2. P<0.05 3. P<0.05 4. P<0.05 5. P>0.05 6. P>0.05 7. P>0.05 |
| Xu ZY 2016 | 50(30/20) 35.2±4.3 | 50(32/18) 35.1 ±4.1 | SYKF tablet (5 tablets, tid) + Losartan Potassium tablet (1 tablets, qd)) | Losartan Potassium tablet (1 tablets, qd) | 1. Effective Rate 2. Adverse Reaction | 1. P<0.05 2. P>0.05 |
| Yan H 2018 | 30(18/12) 52 ±3 | 30(1317) 52±3 | SYKF tablet (5 tablets, tid) + Losartan Potassium tablet (1 tablets, qd)) | Losartan Potassium tablet (1 tablets, qd) | 1. Effective Rate 2. 24-hour urinary protein quantity 3. BUN 4. Scr | 1. P<0.05 2. P<0.05 3. P>0.05 4. P>0.05 |
| Yu GA 2017 | 46(30/16) 46.5±3.8 | 44(26/18) 47.3 ±4.1 | SYKF tablet (5 tablets, tid) + Losartan Potassium tablet (1 tablets, qd)) | Losartan Potassium tablet (1 tablets, qd) | 1. 24-hour urinary protein quantity 2. IL-6,8 | 1. P<0.05 2. P<0.05 |
| Zhao D 2017 | 49(29/20) 42 ±6 | 49(28/21) 42 ±5 | SYKF tablet (5 tablets, tid) + Losartan Potassium tablet (1 tablets, qd)) | Losartan Potassium tablet (1 tablets, qd) | 1. Effective rate 2. Urine NAG enzyme 3. 24-hour urinary protein quantity 4. BUN 5. Scr 6. Systolic blood pressure | 1. P<0.05 2. P<0.05 3. P<0.05 4. P<0.05 5. P<0.05 6. P<0.05 |
| Zheng BL 2014 | 41 | 41 | SYKF tablet (5 tablets, tid) + Losartan Potassium tablet (1 tablets, qd)) | Losartan Potassium tablet (1 tablets, qd) | 1. Effective rate 2. Urine NAG enzyme 3. 24-hour urinary protein quantity 4. IL-6,8 5. BUN 6. Scr | 1. P<0.05 2. P<0.05 3. P<0.05 4. P<0.05 5. P>0.05 6. P>0.05 |

SYKF tablet: Shenyankangfu tablet Scr: Serum creatinine GFR:Glomerular filtration rate BUN: Blood urea nitrogen

shown in Fig 6 and S3 Table in S1 File. The value of BUN did not change significantly in the study group compared to the control group [SMD = -0.42, 95% CI (-0.85, 0), P = 0.05].

**3.3.5 Urine NAG enzyme.** 5 studies [6, 8, 13, 21, 22] reported changes in urine NAG enzyme, and there was statistical heterogeneity between the studies (P = 0.04, $I^2$ = 60%). The random effects model was used for the meta-analysis. The urine NAG enzyme was

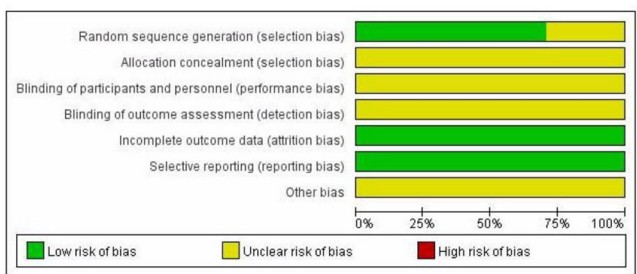

Fig 2. Risk of bias. (A): Risk of bias graph indicating the review authors' rating regarding the risk of bias, presented as percentages, across all of the included studies; (B):Risk of bias summary indicating the review authors' judgments on each risk of bias item for each included study. Green color, low risk of bias; yellow color, unclear risk of bias; red color, high risk of bias.

significantly lower in the experimental group than in the control group [SMD = -0.99, 95% CI (-1.27, -0.72), P < 0.00001]. The results are shown in Fig 7 and S4 Table in S1 File.

**3.3.6 Leukotactin-1 (LKN-1).** 5 studies [6, 7, 15–17] reported changes in leukotactin-1, and there was statistical heterogeneity between the studies (P < 0.00001, I² = 95%). The

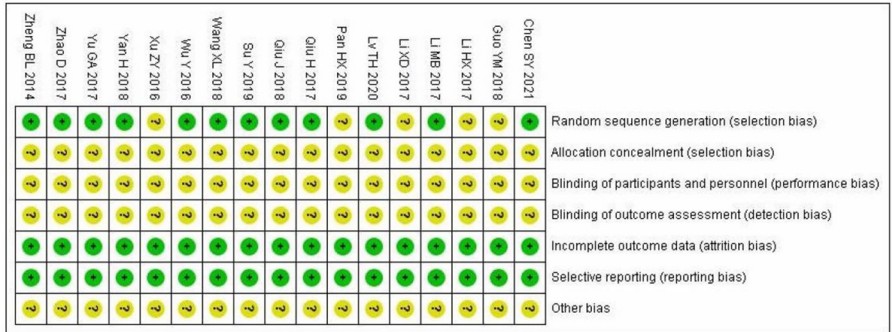

Fig 3. Forest plot of effects of Shenyankangfu tablets and Losartan potassium in the treatment of chronic glomerulonephritis on effective rate.

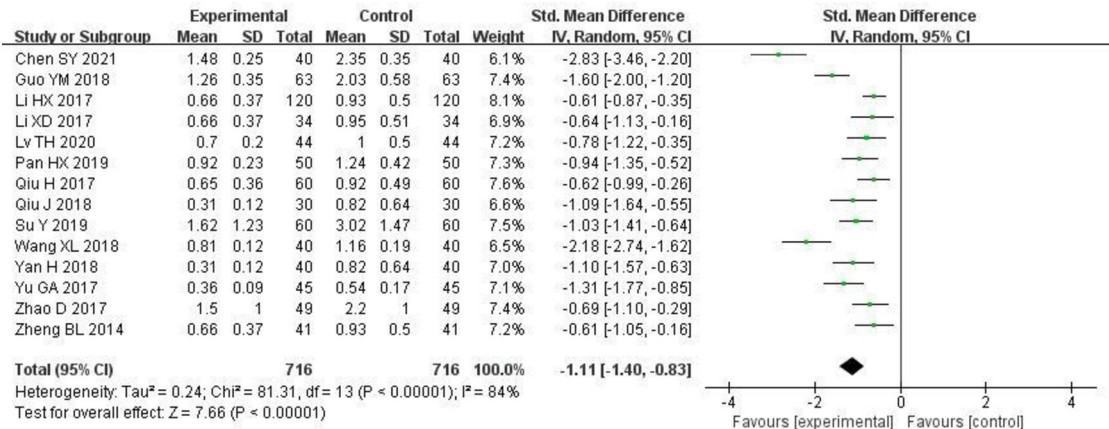

**Fig 4. Forest plot of effects of Shenyankangfu tablets and Losartan potassium in the treatment of chronic glomerulonephritis on 24-hour urine protein quantification.**

random effects model was used for the meta-analysis. The experimental group had a significantly lower LKN-1 than the control group [SMD = -2.43, 95% CI (-3.50, -1.35), P < 0.00001]. The results are shown in Fig 8 and S5 Table in S1 File.

**3.3.7 Adverse reaction.** 6 studies [6, 8, 9, 11–13] reported changes in the rate of adverse reactions. There was no statistical heterogeneity between the studies (P = 0.69, I² = 0%), and the fixed-effect model was used for the meta-analysis. The meta-analysis results showed that the rate of adverse reaction in patients in the study group was significantly lower than that in the control group [RR = 0.43, 95% CI (0.28, 0.66), P < 0.00001]. The results are shown in Fig 9.

**3.3.8 Analysis of publication bias.** The effective rate was used as an indicator to draw an inverted funnel chart, as shown in Fig 10. The scatter points of each study are within the scope of the inverted funnel chart, and the distribution is symmetrical. It suggests that the possibility of publication bias in this study is low.

## 4. Discussion

CGN is a primary glomerular disease characterized by diffuse inflammation and limited glomerular fibrosis with clinical manifestations of proteinuria, hematuria, hypertension, and

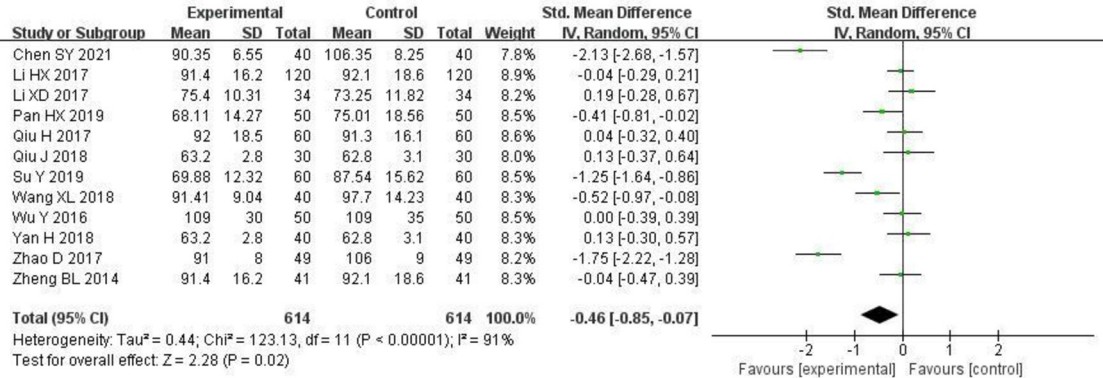

**Fig 5. Forest plot of effects of Shenyankangfu tablets and Losartan potassium in the treatment of chronic glomerulonephritis on Serum creatinine.**

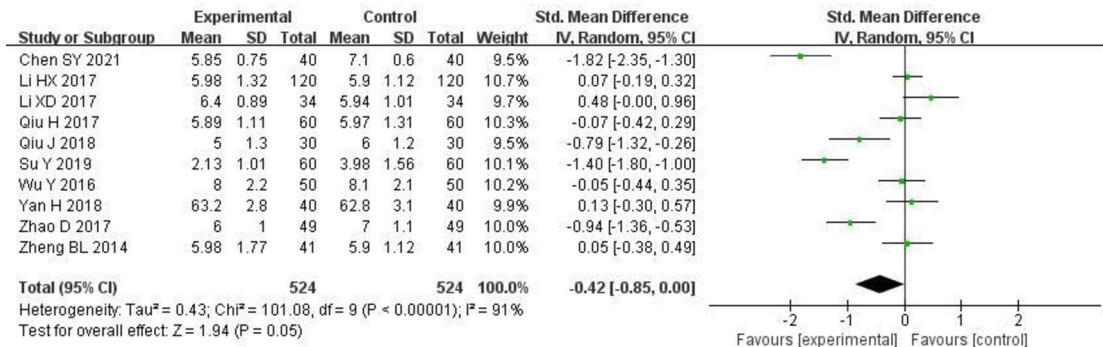

**Fig 6. Forest plot of effects of Shenyankangfu tablets and Losartan potassium in the treatment of chronic glomerulonephritis on blood urea nitrogen.**

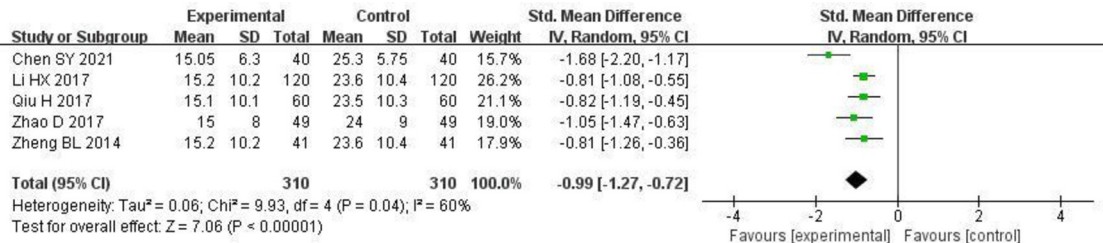

**Fig 7. Forest plot of effects of Shenyankangfu tablets and Losartan potassium in the treatment of chronic glomerulonephritis on urine NAG enzyme.**

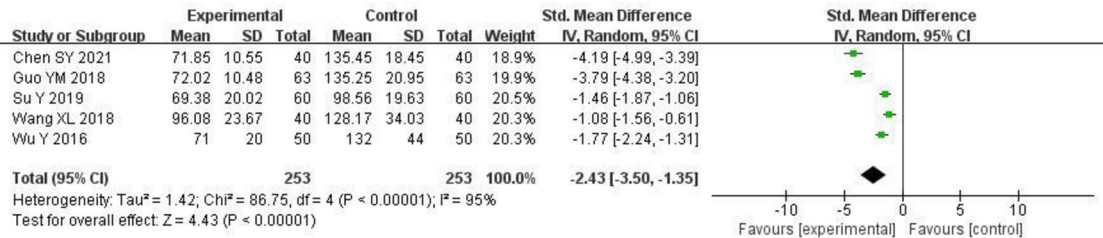

**Fig 8. Forest plot of effects of Shenyankangfu tablets and Losartan potassium in the treatment of chronic glomerulonephritis on leukotactin-1.**

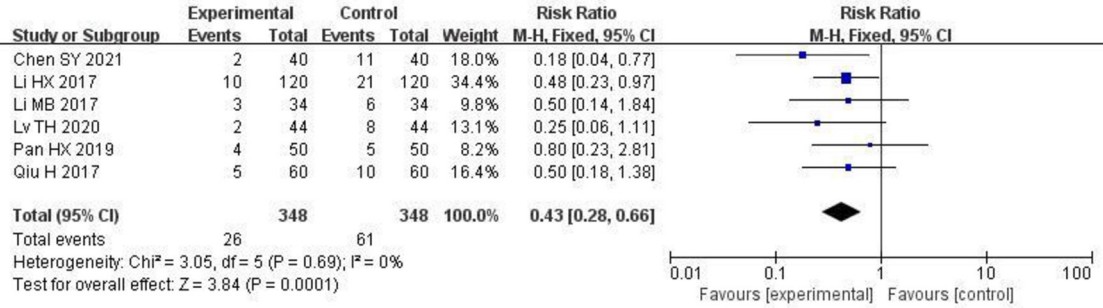

**Fig 9. Forest plot of effects of Shenyankangfu tablets and Losartan potassium in the treatment of chronic glomerulonephritis on adverse reaction.**

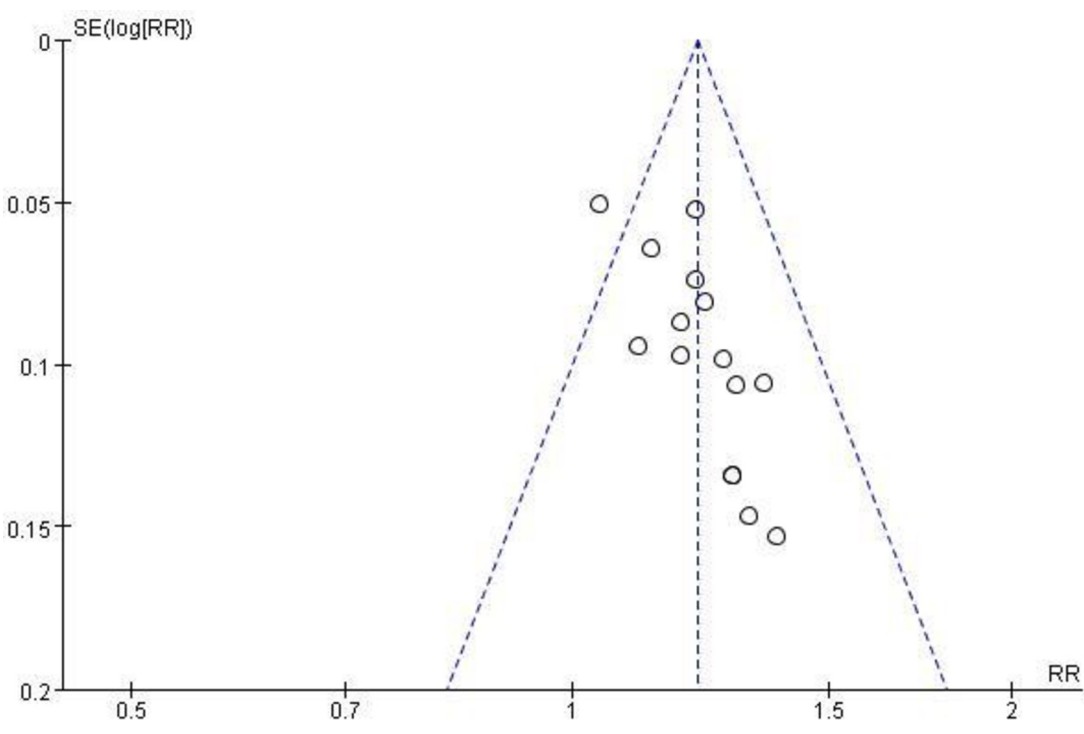

**Fig 10. Inverted funnel plots of effective rate.**

varying degrees of edema. Since CGN is prone to recurrent episodes, if the disease progresses and is not effectively treated, it can result in life-threatening complications such as renal failure, severely affecting the patient's health and quality of life. Although the pathogenesis of chronic glomerulonephritis is not completely understood, most clinical research scholars believe that it is closely related to humoral immunity. When chronic glomerulonephritis is taken into account, the main goal of clinical treatment is to reduce proteinuria.

Losartan potassium, which is an angiotensin II receptor antagonist can selectively compete for angiotensin receptor 2 binding that dilates renal arteries and blood vessels throughout the body, reducing proteinuria and blood pressure [20]. Following treatment, the patient's glomerular hypermetabolic state significantly improves, thereby protecting renal function and reducing the degree of renal function damage. However, it was found during clinical trials that long-term use affects blood pressure, increases the burden on the kidneys, and causes a variety of adverse reactions. In addition, the patient is prone to recurring attacks after discontinuing the drug, which affects the patient's treatment effect.

Modern pharmacological studies have found that Shenyankangfu tablets have anti-inflammatory properties, promote liver protein synthesis, prevent protein loss, regulate immune function, and lower blood pressure [23]. Considering that GN is primarily associated with infection or autoimmune abnormalities, SYKFT tablets may have therapeutic effects by regulating pathways involved in immunity, inflammation, and oxidative stress [24], thereby reducing the occurrence of proteinuria, and hematuria, and delaying the deterioration of renal function. In addition, treatment of CKD relies on effective control of proteinuria and protection of kidney function. In previous research, SYKFT was shown to provide a greater advantage in improving CKD clinical symptoms, controlling proteinuria, protecting kidney function, and reducing Western medicine side effects [25].

A meta-analysis was conducted to assess the efficacy and safety of the combination of Shenyankangfu tablets and losartan potassium in the treatment of chronic glomerulonephritis. The results showed that using Shenyankangfu tablets in combination with losartan potassium significantly improved the total effective rate of the patients in the study group. The 24-hour urine protein, NAG enzyme, and LKN-1 levels were significantly lower in the study group when compared to the control group. Scr and BUN were reduced to varying degrees in both groups of patients, but the difference was insignificant. All raw data could be obtained in study's minimal underlying data in supporting information. The overall incidence of adverse reactions was lower in the study group than in the control group. This indicates that treating chronic glomerulonephritis with the combination of Sheyanyankangfu tablets and losartan potassium is safer. At the same time, it alleviates disease-related symptoms, reduces the influence of cytokine levels, and has fewer adverse reactions, making it more conducive to disease recovery.

This study has the following limitations: 1. The random method of some studies is not clearly stated, and there is no mention of double-blind, which diminishes the reliability of the evidence and may introduce selection bias; 2. The literature is composed solely of Chinese sources, which may result in publication bias. Additional multi-center, randomized, controlled trials with large sample sizes must be conducted to confirm the findings.

## Supporting information

**S1 Checklist. PRISMA 2020 checklist.**
(DOCX)

**S1 Data. Study's minimal underlying data: Raw data involved in meta-analysis.**
(XLSX)

**S1 File. This file consists of the supporting tables for this submission.**
(DOCX)

**S2 File. Supplementary material (abstract): Abstracts of the literature used in the meta-analysis.**
(DOCX)

## Author Contributions

**Conceptualization:** Long-Xun Zhu.

**Data curation:** Pan-Feng Feng, Xiang-Fan Chen, Nan Sheng.

**Funding acquisition:** Xiang-Fan Chen, Nan Sheng.

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
