## [Decision Letter · Decision Letter 0]

6 Jul 2022

PONE-D-21-37134

Meta-analysis of the effectiveness and safety of Shenyankangfu tablets combined with losartan potassium in the treatment of Chronic glomerulonephritis

PLOS ONE

Dear Dr. Zhu,

Thank you for submitting your manuscript to PLOS ONE. After careful consideration, we feel that it has merit but does not fully meet PLOS ONE’s publication criteria as it currently stands. Therefore, we invite you to submit a revised version of the manuscript that addresses the points raised during the review process.

The manuscript focuses on a topic of potential interest. However, the study has major pitfalls that should be addressed to support the conclusion. To mention few of them, i) need to elaborate on the mechanism of action of the drug, as it is not known in western medicine world; ii) need to be more specific on the losartan mechanisms of action; iii) concern about the presentation  of Table 1, where intergroup difference is hard to read and is not self-explanatory; iv) need to add in the limitations that, since the study was conducted in Chinese population, this drug needs to be studied in other populations before it can become mainstream; v) concern about the fact that many of the studies cited are unavailable to non-Chinese readers and this is a problem for making context of this meta-analysis. Would it be possible to have at least the abstract of these studies translated in English to be added as supplementary material?; vii) unclear why they kept mentioning chronic glomerulonephritis as the most common glomerular disease worldwide. The introduction and early parts of the discussion should be re-written to reflect the actual report of worldwide literature on glomerular diseases; viii) need to clarify how chronic glomerulonephritis was defined in the included studies; ix) concern about the fact that the results presented in the abstract and section 3.3.1 of the Results seem to say the opposite of those reported in Figure 3; x) need to provide kidney function reported in the studies included; xi) need to support with references several statements made in the discussion; xii) need to improve the manuscript by a more detailed discussion into the main findings including, for instance, why the control group did better than the main group (Figure 3); xiii) need to provide the number (and percentage) of studies that were low-, moderate-, and high-risk of bias; xiv) need to provide the dose of losartan in the included studies.

We look forward to receiving your revised manuscript.

Kind regards,

Giuseppe Remuzzi

Academic Editor

PLOS ONE

Journal Requirements:

2. Thank you for stating the following financial disclosure: "Nantong Health Commission Fund (No. QA2021007, QA2021006, QA2021014)."

Please state what role the funders took in the study.  If the funders had no role, please state: "The funders had no role in study design, data collection and analysis, decision to publish, or preparation of the manuscript.

Reviewers' comments:

Reviewer's Responses to Questions

**Comments to the Author**

1. Is the manuscript technically sound, and do the data support the conclusions?

Reviewer #1: Yes

Reviewer #2: Partly

2. Has the statistical analysis been performed appropriately and rigorously? 

Reviewer #1: Yes

Reviewer #2: Yes

3. Have the authors made all data underlying the findings in their manuscript fully available?

Reviewer #1: Yes

Reviewer #2: Yes

4. Is the manuscript presented in an intelligible fashion and written in standard English?

Reviewer #1: No

Reviewer #2: Yes

5. Review Comments to the Author

Reviewer #1: Feng et al. conducted meta-analysis of the effectiveness and safety of Shenyankangfu tablets combined with losartan potassium in the treatment of Chronic glomerulonephritis. The results showed that the total effective rate of patients in the study group was significantly improved after using the Shenyankangfu tablets in combination with losartan potassium. 24-hour urine protein, urine NAG enzyme, LKN-1 are lower than the control group. BUN and Cr were not very different between the groups and adverse reactions were less frequent in the study group.

1. Would recommend elaborating on the mechanism of action of the drug as it is not known in western medicine world. Also I would recommend being more specific in losartan's MOA (decrease glomerular hypertension from efferent dilatation, some direct podocyte effects and less fibrosis through TGF-B).

2.Table 1 intergroup difference is hard to read and is not self explanatory.

3. Would check for spelling and grammatical errors.

4. Since the study was conducted in Chinese population, this drug needs to be studied in other population before it can become mainstream.

Reviewer #2: Feng et al have conducted this meta-analysis on the effectiveness and safety of Shenyankangfu tablet combined with Losartan potassium in the treatment of chronic glomerulonephritis (CGN). Please see my comments on this manuscript:

MAJOR:

1. My main problem with this paper submitted to PLOS ONE with largely English readership (and the authors have listed this as a limitation to this study) is the inclusion of only Chinese studies (or studies published in Chinese literature). Many of the studies cited are unavailable to non-Chinese readers and this is a problem for making context of this meta-analysis.

2. It is unclear to me why the authors kept mentioning CGN as the most common glomerular disease worldwide. This is not substantiated in any report of worldwide literature on glomerular diseases. IgAN is the commonest GN in European and Asian countries (including China - See Nephrol Dial Transplant. 2010 Feb;25(2):334-6.) while MCD / FSGS is common in Latin Americans and Africa (see Nephrol Dial Transplant. 2010 Feb;25(2):490-6; and PLoS One. 2016 Mar 24;11(3):e0152203.). CGN represents the final common pathway of most glomerular disorders (primary or secondary) if not well treated or does not respond to therapies and is characterized by interstitial fibrosis, glomerulosclerosis and tubular atrophy. The introduction and early parts of the discussion should therefore be re-written to reflect this.

3. How was chronic glomerulonephritis defined in the included studies? Was this from the same type of primary or secondary GN?

4. Figure 3 clearly shows that the "Total effective rate" was higher (favors) the control group. The results you presented in the abstract and section 3.3.1 of the results seems to say the opposite. This should be clarified.

5. It is unclear about kidney function of the studies included. I think the authors should at least summarize baseline and end-of study serum creatinine, eGFR and ACR (and other tests) in a Table (main or supplementary) to improve context and understanding of the included studies.

6. Several statements were made in the discussion that were not backed up with references (e.g. "Modern pharmacological studies have found that Shenyankangfu tablets have the functions of anti-inflammatory, promoting liver protein synthesis, preventing protein loss, regulating immune function, and lowering blood pressure." Can you provide references to back up this statement as well as others?"

7. I think the study can be improved by a more detailed discussion into the main findings including, for instance, why the control group did better than the main group (Figure 3) amongst others to be discussed.

8. Based on the risk of bias analysis carried out, please clearly provide the number (and percentage) of studies that were low-, moderate- and high-risk of bias.

9. Did the included studies provide dose of Losartan? This should be provided as it has implications on the results provided.

MINOR

1. Please expand the abbreviations before first use (e.g. SMD; urine NAG, etc)

2. Provide the PROSPERO registration number for this study protocol

6. PLOS authors have the option to publish the peer review history of their article (what does this mean?). If published, this will include your full peer review and any attached files.

Reviewer #1: No

Reviewer #2: **Yes: **Ikechi Okpechi

---

## [Author Response · Author response to Decision Letter 0]

2 Aug 2022

Reviewer #1: Feng et al. conducted meta-analysis of the effectiveness and safety of Shenyankangfu tablets combined with losartan potassium in the treatment of Chronic glomerulonephritis. The results showed that the total effective rate of patients in the study group was significantly improved after using the Shenyankangfu tablets in combination with losartan potassium. 24-hour urine protein, urine NAG enzyme, LKN-1 are lower than the control group. BUN and Cr were not very different between the groups and adverse reactions were less frequent in the study group.

1.Would recommend elaborating on the mechanism of action of the drug as it is not known in western medicine world. Also I would recommend being more specific in losartan's MOA (decrease glomerular hypertension from efferent dilatation, some direct podocyte effects and less fibrosis through TGF-B).

Response: We sincerely appreciate the valuable comments. the Shenyankangfu Tablet (SYKFT) treatment of GN primarily involved infection, inflammation, or immunity processes. Considering that GN was primarily correlated with infection or autoimmune abnormalities, SYKFT may exert therapeutic effects by regulating pathways involved in immunity, inflammation, and oxidative stress [1]. We have added the mechanism of action of SYKFT in the section of discussion in the manuscript. The aim of this study was to systematically review the efficacy and safety of Shenyankangfu tablets combined with losartan potassium in the treatment of chronic glomerulonephritis. Our next research will focus on losartan's MOA. 

1.Jin M, Ren W, Zhang W, et al. Exploring the Underlying Mechanism of Shenyankangfu Tablet in the Treatment of Glomerulonephritis Through Network Pharmacology, Machine Learning, Molecular Docking, and Experimental Validation. Drug Des Devel Ther. 2021;15:4585-4601. 

2.Table 1 intergroup difference is hard to read and is not self explanatory.

Response: Thanks for your professional suggestion. We have made adjustments to the way we write of Table 1 intergroup difference. The numbers in intergroup difference are in one-to-one correspondence with the numbers in outcome index.

3. Would check for spelling and grammatical errors.

Response: Thanks for your professional suggestion. We have made corrections to spelling and grammar mistakes in the manuscript.

4.Since the study was conducted in Chinese population, this drug needs to be studied in other population before it can become mainstream.

Response: Thanks for this suggestion. Shenyankangfu Tablet (SYKFT) is a Chinese patent medicine that has been used widely to decrease proteinuria and the progression of chronic kidney disease. Its safety has been confirmed by a multicenter randomized controlled trial [1]. Of course, the next step is to conduct research in other population to make it mainstream

1.Wu J, Duan SW, Yang HT, et al. Efficacy and safety of Shenyankangfu Tablet, a Chinese patent medicine, for primary glomerulonephritis: A multicenter randomized controlled trial. J Integr Med. 2021;1:009.

Reviewer #2: Feng et al have conducted this meta-analysis on the effectiveness and safety of Shenyankangfu tablet combined with Losartan potassium in the treatment of chronic glomerulonephritis (CGN). Please see my comments on this manuscript:

MAJOR:

1. My main problem with this paper submitted to PLOS ONE with largely English readership (and the authors have listed this as a limitation to this study) is the inclusion of only Chinese studies (or studies published in Chinese literature). Many of the studies cited are unavailable to non-Chinese readers and this is a problem for making context of this meta-analysis.

Response: Thank you for this constructive suggestion. We have provided the abstract of these studies translated in English as supplementary material (abstract).

2. It is unclear to me why the authors kept mentioning CGN as the most common glomerular disease worldwide. This is not substantiated in any report of worldwide literature on glomerular diseases. IgAN is the commonest GN in European and Asian countries (including China - See Nephrol Dial Transplant. 2010 Feb;25(2):334-6.) while MCD / FSGS is common in Latin Americans and Africa (see Nephrol Dial Transplant. 2010 Feb;25(2):490-6; and PLoS One. 2016 Mar 24;11(3):e0152203.). CGN represents the final common pathway of most glomerular disorders (primary or secondary) if not well treated or does not respond to therapies and is characterized by interstitial fibrosis, glomerulosclerosis and tubular atrophy. The introduction and early parts of the discussion should therefore be re-written to reflect this.

Response: Thank you for this constructive suggestion. We have re-written the introduction and early parts of the discussion.

3. How was chronic glomerulonephritis defined in the included studies? Was this from the same type of primary or secondary GN?

Response: Thanks for your professional suggestion. Chronic glomerulonephritis is mainly a disease with clinical symptoms such as renal dysfunction, proteinuria, hematuria, hypertension and other clinical symptoms caused by various etiologies. It was from the same type of primary GN.

4. Figure 3 clearly shows that the "Total effective rate" was higher (favors) the control group. The results you presented in the abstract and section 3.3.1 of the results seems to say the opposite. This should be clarified.

Response: Thanks for your professional suggestion. In Fig.3, the effective rate in experiment group was 92.08% (674/732). In control group, the effective rate was 75.68% (554/732). The experimental group was more effective. The results I presented in the abstract and section 3.3.1 of the results was that the effective rate in experimental group were significantly higher than that of control group. This conclusion is consistent with the results expressed in Fig.3

5.It is unclear about kidney function of the studies included. I think the authors should at least summarize baseline and end-of study serum creatinine, eGFR and ACR (and other tests) in a Table (main or supplementary) to improve context and understanding of the included studies.

Response: Thanks for your professional suggestion. We have summarized baseline and end-of study 24-hour urine protein quantification, Serum creatinine, blood urea nitrogen, Urine NAG enzyme and leukotactin-1 in the supplementary materials (table). 

6. Several statements were made in the discussion that were not backed up with references (e.g. "Modern pharmacological studies have found that Shenyankangfu tablets have the functions of anti-inflammatory, promoting liver protein synthesis, preventing protein loss, regulating immune function, and lowering blood pressure." Can you provide references to back up this statement as well as others?"

Response: Thanks for your professional suggestion. We have added the reference in appropriate place in the manuscript.

7. I think the study can be improved by a more detailed discussion into the main findings including, for instance, why the control group did better than the main group (Figure 3) amongst others to be discussed.

Response: Thanks for your professional suggestion. In Fig.3, the effective rate in experiment group was 92.08% (674/732). In control group, the effective rate was 75.68% (554/732). The experimental group was more effective. The results I presented in the abstract and section 3.3.1 of the results was that the effective rate in experimental group were significantly higher than that of control group. This conclusion is consistent with the results expressed in Fig.3

8. Based on the risk of bias analysis carried out, please clearly provide the number (and percentage) of studies that were low-, moderate- and high-risk of bias.

Response: Thanks for your professional suggestion. The bias risk of included studies was assessed according to the Cochran Handbook. The evaluation content includes: random sequence generation, allocation concealment, blinding of participants and personnel, blinding of outcome assessment, incomplete outcome data, selective reporting and other bias. The low risk, high risk, and unclear risk were used to describle each item. For random sequence generation, there are 12 studies (70.58%) in low risk of bias and 5 studies (29.41%) in unclear risk. As for allocation concealment, blinding of participants and personnel, blinding of outcome assessment and other bias, all studies (100%) were unclear risk. In addition, all studies (100%) were low risk in incomplete outcome data and selective reporting. 

9.Did the included studies provide dose of Losartan? This should be provided as it has implications on the results provided.

Response: Thanks for your professional suggestion. The dose of Losartan was 50mg, Po, qd. We have added the dose of Losartan in Inclusion criteria in the section of Materials and Methods

MINOR

1. Please expand the abbreviations before first use (e.g. SMD; urine NAG, etc)

Response: Thanks for your professional suggestion. SMD: standardized mean difference. NAG:N acetyl β-D Glucosaminidase. We have assed the full name of SMD and NAG in the manuscript.

2. Provide the PROSPERO registration number for this study protocol

Response: Thanks for your suggestion. The PROSPERO registration number for this study protocol is CRD42022345149. We have added the PROSPERO registration number in the section of search strategy in the manuscript.

---

## [Decision Letter · Decision Letter 1]

7 Sep 2022

PONE-D-21-37134R1Meta-analysis of the effectiveness and safety of Shenyankangfu tablets combined with losartan potassium in the treatment of Chronic glomerulonephritisPLOS ONE

Dear Dr. Zhu,

Thank you for submitting your manuscript to PLOS ONE. After careful consideration, we feel that it has merit but does not fully meet PLOS ONE’s publication criteria as it currently stands. Therefore, we invite you to submit a revised version of the manuscript that addresses the points raised during the review process.

**The revised manuscript is improved. However, few points remain to be addressed, namely, i) need to elaborate more on RAAS blockers on lowering intraglomerular hypertension, direct podocyte effects by blocking ATII receptors; ii) need to provide more in depth details about the mechanism of action of Shenyankangfu tablets; iii) need minor revision regarding grammatical errors.**

We look forward to receiving your revised manuscript.

Kind regards,

Giuseppe Remuzzi

Academic Editor

PLOS ONE

Journal Requirements:

Reviewers' comments:

Reviewer's Responses to Questions

**Comments to the Author**

1. If the authors have adequately addressed your comments raised in a previous round of review and you feel that this manuscript is now acceptable for publication, you may indicate that here to bypass the “Comments to the Author” section, enter your conflict of interest statement in the “Confidential to Editor” section, and submit your "Accept" recommendation.

Reviewer #1: All comments have been addressed

Reviewer #2: All comments have been addressed

2. Is the manuscript technically sound, and do the data support the conclusions?

Reviewer #1: Yes

Reviewer #2: Yes

3. Has the statistical analysis been performed appropriately and rigorously? 

Reviewer #1: Yes

Reviewer #2: Yes

4. Have the authors made all data underlying the findings in their manuscript fully available?

Reviewer #1: Yes

Reviewer #2: Yes

5. Is the manuscript presented in an intelligible fashion and written in standard English?

Reviewer #1: No

Reviewer #2: Yes

6. Review Comments to the Author

Reviewer #1: The authors have addressed most of the the comments from the reviewers. I would also recommend elaborating more on RAAS blockers on lowering intraglomerular hypertension, direct podocyte effects by blocking ATII receptors. Also would recommend delving deep in the mechanism of action of Shenyankangfu tablets. The manuscript still needs minor revision regarding grammatical errors.

Reviewer #2: A Meta-analysis of the effectiveness and safety of Shenyankangfu tablets combined with losartan potassium in the treatment of Chronic glomerulonephritis. I have no further comments.

7. PLOS authors have the option to publish the peer review history of their article (what does this mean?). If published, this will include your full peer review and any attached files.

Reviewer #1: No

Reviewer #2: No

---

## [Author Response · Author response to Decision Letter 1]

13 Sep 2022

The revised manuscript is improved. However, few points remain to be addressed, 

namely, i) need to elaborate more on RAAS blockers on lowering intraglomerular hypertension, direct podocyte effects by blocking ATII receptors; 

Response: Thanks for your professional suggestion. Losartan potassium, which is an angiotensin II receptor antagonist can selectively compete for angiotensin receptor 2 binding that dilates renal arteries and blood vessels throughout the body, reducing proteinuria and blood pressure.

ii) need to provide more in depth details about the mechanism of action of Shenyankangfu tablets;

Response: Thanks for your professional suggestion. We have provided more in depth details about the mechanism of action of Shenyankangfu tablets. the Shenyankangfu Tablet (SYKFT) treatment of GN primarily involved infection, inflammation, or immunity processes. Considering that GN was primarily correlated with infection or autoimmune abnormalities, SYKFT may exert therapeutic effects by regulating pathways involved in immunity, inflammation, and oxidative stress [1]. In addition, treatment of CKD relies on effective control of proteinuria and protection of kidney function. In previous research, SYKFT was shown to provide a greater advantage in improving CKD clinical symptoms, controlling proteinuria, protecting kidney function, and reducing Western medicine side effects [2]. 

1.Jin M, Ren W, Zhang W, et al. Exploring the Underlying Mechanism of Shenyankangfu Tablet in the Treatment of Glomerulonephritis Through Network Pharmacology, Machine Learning, Molecular Docking, and Experimental Validation. Drug Des Devel Ther. 2021;15:4585-4601. 

2.Lu H, Tang S, Su B. Clinical observation on treating 50 cases of chronic nephritis by Shenyan Kangfu tablets plus valsartan. Clin J Chin Med. 2012;4:79–80. doi: 10.4236/cm.2013.43012. 

 iii) need minor revision regarding grammatical errors.

Response: We sincerely appreciate the valuable comments. We found a professional language editing company to revise grammatical errors of the manuscript, and the proof is as follows:

---

## [Decision Letter · Decision Letter 2]

22 Sep 2022

Meta-analysis of the effectiveness and safety of Shenyankangfu tablets combined with losartan potassium in the treatment of Chronic glomerulonephritis

PONE-D-21-37134R2

Dear Dr. Zhu,

We’re pleased to inform you that your manuscript has been judged scientifically suitable for publication and will be formally accepted for publication once it meets all outstanding technical requirements.

**The re-revised version of the manuscript is definitely improved. The authors have now properly addressed all the reviewers’ comments and criticisms.**

Kind regards,

Giuseppe Remuzzi

Academic Editor

PLOS ONE

Additional Editor Comments (optional):

Reviewers' comments:

Reviewer's Responses to Questions

**Comments to the Author**

1. If the authors have adequately addressed your comments raised in a previous round of review and you feel that this manuscript is now acceptable for publication, you may indicate that here to bypass the “Comments to the Author” section, enter your conflict of interest statement in the “Confidential to Editor” section, and submit your "Accept" recommendation.

Reviewer #1: All comments have been addressed

Reviewer #2: All comments have been addressed

2. Is the manuscript technically sound, and do the data support the conclusions?

Reviewer #1: Yes

Reviewer #2: Yes

3. Has the statistical analysis been performed appropriately and rigorously? 

Reviewer #1: Yes

Reviewer #2: Yes

4. Have the authors made all data underlying the findings in their manuscript fully available?

Reviewer #1: Yes

Reviewer #2: Yes

5. Is the manuscript presented in an intelligible fashion and written in standard English?

Reviewer #1: Yes

Reviewer #2: Yes

6. Review Comments to the Author

Reviewer #1: Combination of Shenyankangfu tablets combined with losartan potassium in the treatment of Chronic glomerulonephritis alleviates disease-related symptoms, reduces the influence of cytokine levels, and has fewer adverse reactions compared to losartan alone. This needs to be verified in other populations other than Han Chinese.

Reviewer #2: The authors have shown in their paper that it is safer to treat CGN with Shyenyankangfu tablets combined with losartan potassium. They also show that this compound alleviates disease-related symptoms, reduces the influence of cytokine levels, and has fewer adverse reactions. I have no further comments for the authors who have now responded to all queries.

7. PLOS authors have the option to publish the peer review history of their article (what does this mean?). If published, this will include your full peer review and any attached files.

Reviewer #1: No

Reviewer #2: No

---

## [Editor Report · Acceptance letter]

29 Sep 2022

PONE-D-21-37134R2 

Meta-analysis of the effectiveness and safety of Shenyankangfu tablets combined with losartan potassium in the treatment of chronic glomerulonephritis 

Dear Dr. Zhu:

I'm pleased to inform you that your manuscript has been deemed suitable for publication in PLOS ONE. Congratulations! Your manuscript is now with our production department. 

Kind regards, 

on behalf of

Prof. Giuseppe Remuzzi 

Academic Editor

PLOS ONE